# Diagnostic Accuracy of Insulinoma-Associated Protein 1 in Pulmonary Neuroendocrine Carcinomas: A Systematic Review and Meta-Analysis

**DOI:** 10.3390/cancers17152544

**Published:** 2025-07-31

**Authors:** Risa Waki, Saya Haketa, Riona Aburaki, Nobuyuki Horita

**Affiliations:** Chemotherapy Center, Yokohama City University, 3-9 Fukuura, Kanazawa-ku, Yokohama 236-0004, Japan; e213090d@yokohama-cu.ac.jp (R.W.); e213065g@yokohama-cu.ac.jp (S.H.); e203003d@yokohama-cu.ac.jp (R.A.)

**Keywords:** lung neoplasms, neuroendocrine tumors, large cell neuroendocrine carcinoma, small cell lung cancer, INSM1, sensitivity and specificity

## Abstract

Diagnosing neuroendocrine carcinomas can be challenging. Doctors usually need to use a combination of several markers to accurately identify lung cancers of this type. A newer marker, INSM1, has been suggested as a single test that might work just as well on its own. However, results from individual studies have varied. In this study, we carefully reviewed and combined data from 14 previous studies to evaluate how well INSM1 performs in diagnosing pulmonary neuroendocrine carcinomas. We found that INSM1 shows consistently high accuracy, especially in terms of ruling out other cancer types. Our findings suggest that INSM1 could be used as a reliable single marker in lung cancer diagnosis, which may simplify testing and improve consistency in pathology laboratories.

## 1. Introduction

Lung neoplasms encompass a diverse spectrum of malignancies, each with distinct histological features, clinical behaviors, and therapeutic approaches [1,2]. Among lung neoplasms, small cell lung cancer (SCLC) is distinguished by its aggressive proliferation and rapid metastatic potential, leading to a markedly poor prognosis compared to other lung cancer subtypes [3,4,5]. At the time of diagnosis, SCLC is frequently inoperable, and systemic chemotherapy remains the primary treatment modality [2,6].

Pulmonary large cell neuroendocrine carcinoma (LCNEC) and SCLC are two types of high-grade neuroendocrine carcinomas of the lung with poor prognoses, comprising approximately 2–3% and 15% of all lung cancers, respectively [7,8]. LCNEC, though historically classified as a subtype of non-small cell lung cancer (NSCLC), shares a wide range of clinical, biological, and therapeutic characteristics with SCLC [4]. Consequently, accurate differentiation of LCNEC and SCLC from other NSCLC subtypes—such as adenocarcinoma, squamous cell carcinoma, and conventional large cell carcinoma lacking neuroendocrine features—is essential for appropriate prognostication and patient management [9,10,11].

LCNEC and SCLC, along with typical and atypical carcinoid tumors, are collectively classified as pulmonary neuroendocrine neoplasms [12]. While the diagnosis of LCNEC requires confirmation of neuroendocrine (NE) differentiation by immunohistochemical (IHC) staining with NE markers, the diagnosis of SCLC is based primarily on morphological criteria—specifically, small, round to fusiform tumor cells with scant cytoplasm, finely granular hyperchromatic nuclei, and inconspicuous or absent nucleoli [12,13]. However, in practice, morphological evaluation alone may be insufficient in poorly preserved or ambiguous specimens. In such cases, confirmation of NE differentiation by IHC can improve diagnostic reliability [2].

The most commonly used NE markers include chromogranin A (UniProtKB: P10645), synaptophysin (P08247), and CD56 (P13591) [12]. Synaptophysin and CD56 generally exhibit higher sensitivity but lower specificity, while chromogranin A tends to be more specific but less sensitive [14]. Due to these limitations, these markers are typically used in combination to increase diagnostic accuracy [15]. Recent WHO best practice recommendations support this panel approach and suggest that any degree of positivity be considered sufficient for diagnosis [16]. However, even with panel-based testing, diagnostic accuracy remains suboptimal, and some NE tumors may test negative for all three markers [12,17]. Furthermore, their cytoplasmic staining patterns can be difficult to interpret, especially when staining is weak or focal, sometimes leading to confusion with nonspecific background signals [18].

In recent years, researchers have explored the use of insulinoma-associated protein 1 (INSM1) as a novel neuroendocrine marker [11,19]. INSM1 (UniProtKB: Q01101) is a zinc-finger transcription factor that plays a critical role in the development of neuroendocrine tissues [20]. It encodes a 510-amino acid protein containing five zinc finger motifs at the carboxyl terminus [21,22]. INSM1 was initially identified as a complementary DNA clone (IA-1) in a human pancreatic insulinoma subtraction library in 1992 by Goto et al. and was shown to be expressed in human and murine insulinoma tissues [20,23].

Subsequent studies demonstrated INSM1 expression in SCLC, LCNEC, and carcinoid tumors, but not in NSCLC lacking NE differentiation [19,24]. As a nuclear marker, INSM1 offers advantages in interpretability over traditional cytoplasmic NE markers [18]. INSM1 has since been applied as a diagnostic marker for neuroendocrine neoplasms in various organs, including the lung, pancreas, and prostate [14,25,26].

Recent studies have suggested that INSM1 has high diagnostic accuracy in identifying pulmonary neuroendocrine neoplasms, particularly SCLC and LCNEC, with reported sensitivity ranging from 68% to 95% and specificity from 95% to 99% [11,19]. However, reported estimates of sensitivity and specificity vary across studies, highlighting the need for a comprehensive evaluation of INSM1′s performance.

To address this gap, the present study aims to conduct a systematic review and meta-analysis to assess the diagnostic accuracy of using INSM1 to identify SCLC and LCNEC. This analysis seeks to synthesize the existing evidence, clarify diagnostic performance, and inform clinical practice regarding the utility of INSM1 in the pathological classification of pulmonary neuroendocrine carcinomas (NECs).

## 2. Methods

### 2.1. Study Overview

This systematic review and meta-analysis was conducted in accordance with the Preferred Reporting Items for Systematic Reviews and Meta-Analyses of Diagnostic Test Accuracy Studies (PRISMA-DTA) guidelines [27]. The protocol was registered on the University Hospital Medical Information Network Clinical Trials Registration website (UMIN000056564). According to Yokohama City University, Institutional Review Board approval was deemed unnecessary due to the study’s nature.

### 2.2. Study Search

Three major online databases, PubMed, Web of Science, and Embase, were systematically searched on 25 December 2024. The PubMed search strategy was as follows: (insulinoma-associated protein 1 OR INSM1 OR INSM-1) AND (lung OR pulmonary OR respiratory OR thoracic OR bronchial OR bronchogenic OR tracheal OR alveolar). Details of the search strategy are provided in Appendix A.

Two independent reviewers (RW and AH) screened titles and abstracts and assessed the full texts to identify eligible studies. Discrepancies were resolved through discussion. Additionally, reference lists of review articles and included studies were manually searched to identify further relevant studies.

### 2.3. Study Selection

Eligible studies included full-text articles, brief reports, and conference abstracts published in English that reported sensitivity and specificity data for the immunohistochemical marker INSM1 in diagnosing SCLC and LCNEC. We excluded studies focusing on low-grade neuroendocrine tumors such as typical or atypical carcinoids. We excluded articles providing only sensitivity or specificity data, as bivariate analysis requires both. We included studies that evaluated the diagnostic performance of INSM1 for high-grade neuroendocrine carcinomas (i.e., SCLC and LCNEC) either alone or in direct comparison with other neuroendocrine markers (e.g., Chromogranin A, Synaptophysin, CD56), although we extracted and analyzed only diagnostic performance data (sensitivity and specificity) specific to INSM1. Although one-gate patient recruitment is preferable for diagnostic accuracy assessments, two-gate studies—e.g., those that selected LCNEC patients and control groups (such as adenocarcinoma or squamous cell carcinoma) from separate datasets—were also included.

All sample types, including surgical specimens, bronchoscopic specimens, and pleural effusion cell blocks, were eligible. Additionally, samples from non-pulmonary origins were accepted if they represented metastatic lung cancer in other organs or lymph nodes. However, studies focusing on neuroendocrine neoplasms or other non-pulmonary neuroendocrine neoplasms (e.g., Merkel cell carcinoma) that did not provide lung cancer-specific data were excluded, even if INSM1 data were available.

The reference standard was a final pathological diagnosis based on routine histopathological evaluation by experienced pathologists. Although not all studies explicitly reported the diagnostic criteria, most were based on the WHO classification and conventional immunohistochemical markers.

### 2.4. Risk of Bias

The quality of the included studies and the risk of bias were assessed using the QUADAS-2 tool [28]. QUADAS-2 is a standardized tool used to evaluate the risk of bias and applicability of primary diagnostic accuracy studies. It examines four domains: patient selection, index test, reference standard, and flow and timing. Each domain is rated for risk of bias and concerns regarding applicability.

In this meta-analysis, the results of the QUADAS-2 assessment were used descriptively to summarize the methodological quality of the included studies. The QUADAS-2 ratings did not influence the inclusion or exclusion of studies, nor were they used to apply statistical weights or perform subgroup analyses. However, we considered potential sources of bias identified in the QUADAS-2 evaluation when interpreting the results of the meta-analysis.

### 2.5. Outcomes

Two analytical models were employed (Table 1):

The NSCLC model was designed to differentiate LCNEC from other non-small cell lung cancers.

The lung cancer model aimed to distinguish NECs (specifically SCLC and LCNEC) from non-neuroendocrine (non-NE) lung cancers.

Sensitivity, specificity, and the area under the curve (AUC) were evaluated. In addition, positive likelihood ratio (PLR), negative likelihood ratio (NLR), and diagnostic odds ratio (DOR) were calculated to further summarize diagnostic performance. If multiple cutoffs were used in an original article, all weakly, moderately, and strongly positive results were collectively considered positive. Thresholds for INSM1 positivity varied across the included studies, ranging from ≥1% to ≥8.68% nuclear staining, and some studies used H-score–based criteria. Due to this heterogeneity and lack of consistent reporting, a pooled threshold could not be calculated. Instead, all cases that were interpreted as INSM1-positive by the original study authors were treated as positive in our analysis.

**Table 1 cancers-17-02544-t001:** Study models.

	SCLC	NSCLC	
		LCNEC	NSCLC Other Than LCNEC
NSCLC model	excluded	positive	negative
Lung cancer model	positive	positive	negative

The figure outlines the two diagnostic models assessed in this study. In the NSCLC model, the positive group consisted of LCNEC cases, while the negative group included other non-small cell lung cancers (NSCLC other than LCNEC). In the lung cancer model, the positive group included both SCLC and LCNEC cases, and the negative group consisted of NSCLC cases other than LCNEC. Dark gray shading indicates the positive group, and light gray shading indicates the negative group. Abbreviations: SCLC, small cell lung cancer; NSCLC, non-small cell lung cancer; LCNEC, large cell neuroendocrine carcinoma.

### 2.6. Use of Generative AI Tools

Portions of this manuscript, including phrasing and paragraph structure, were refined using a generative AI tool (ChatGPT-4o, OpenAI). The authors reviewed and edited all AI-assisted content to ensure accuracy, originality, and scientific integrity.

### 2.7. Data Extraction

Two reviewers (RW and AH) independently extracted the following data: first author’s name, publication year, country, the number of patients with positive results, the total number of patients evaluated, and information related to the QUADAS-2 assessment.

### 2.8. Statistical Analysis

We used Meta-DiSc version 1.4 (Informer Technologies, Inc.) [29] with a bivariate model to calculate pooled sensitivity and specificity and to construct a summary receiver operating characteristic (SROC) curve. The bivariate analysis is a statistical approach commonly used in meta-analyses of diagnostic tests. It allows simultaneous modeling of sensitivity and specificity while accounting for the potential correlation between them across studies. This method improves the accuracy of pooled estimates and supports SROC curve construction [30]. We also calculated the pooled diagnostic odds ratio (DOR) with corresponding 95% confidence intervals using Meta-DiSc version 1.4 (Informer Technologies, Inc.) [29] to summarize the overall diagnostic performance of INSM1. DOR combines sensitivity and specificity into a single indicator and provides a useful single-number measure of diagnostic performance. Additionally, we calculated pooled positive likelihood ratio (PLR) and negative likelihood ratio (NLR) with 95% confidence intervals using Meta-DiSc version 1.4 (Informer Technologies, Inc.) [29].

As a sensitivity analysis, the bivariate random-effects model was also applied using the “reitsma” command in the “mada” R package (version 0.5.10) in R (version 4.0.4) [30]. AUC values were interpreted as follows: ≥0.97, excellent; 0.93–0.96, very good; 0.75–0.92, good; and 0.5–0.74, fair [31]. Statistical significance was defined as *p* < 0.05.

To assess potential threshold effects, we calculated the Spearman correlation coefficient between sensitivity and 1 − specificity across studies using Excel Toukei (BellCurve. Social Survey Research Information, Tokyo, Japan). A threshold effect was considered present when the correlation coefficient (r) was ≥0.60 with a *p*-value < 0.05, reflecting a positive correlation consistent with a trade-off between sensitivity and specificity. This approach is supported by previous literature, which indicates that a threshold effect typically produces a positive correlation between sensitivity and 1 − specificity, or equivalently, a negative correlation between sensitivity and specificity [32]. Heterogeneity was assessed using I^2^ statistics calculated in Meta-DiSc version 1.4 (Informer Technologies, Inc.) and interpreted as follows: I^2^ = 0%, no heterogeneity; 0% < I^2^ < 25%, minimal heterogeneity; 25% ≤ I^2^ < 50%, mild heterogeneity; 50% ≤ I^2^ < 75%, moderate heterogeneity; and I^2^ ≥ 75%, strong heterogeneity [33].

All statistical analyses were performed using Meta-DiSc version 1.4 (Informer Technologies, Inc., Madrid, Spain), R version 4.0.4 with the mada package version 0.5.10, and Excel Toukei (BellCurve. Social Survey Research Information, Tokyo, Japan).

## 3. Results

### 3.1. Study Search and Study Characteristics

A total of 521 records were identified through database searching (n = 518) and hand searching (n = 3). (Figure 1) After removing duplicates, performing title and abstract screening, and reviewing full texts, 14 articles were included in the quantitative and qualitative synthesis [10,11,18,19,34,35,36,37,38,39,40,41,42,43]. The publication years ranged from 2015 to 2024. Geographically, eight studies were conducted in the United States, three in Japan, one in India, one in Sweden, and one in Germany. Of the included studies, 11 were full-length original articles and three were conference abstracts; all were published in English.

A total of 521 records were identified (518 through database searching and 3 through hand searching). After removing duplicates, screening titles and abstracts, and assessing full texts, 14 studies were included in the final quantitative and qualitative synthesis. PRISMA, Preferred Reporting Items for Systematic reviews and Meta-Analyses; WOS, Web of Science.

In total, 3218 specimens were analyzed across the included studies. The specimen types varied and included surgical specimens, pleural effusion cell blocks, bronchoscopic biopsies, cytology specimens, and cytology smears. One study did not specify the specimen type (Table 2). Duplicated samples from the same individuals were excluded. Two studies included cases with mixed tumor types; in these, INSM1 positivity was assessed and reported separately for each component. The cut-off criteria for INSM1 positivity varied among the studies. Specifically, two studies considered any degree of nuclear staining as positive. All but two studies used the INSM1 antibody clone sc-271408 (Santa Cruz Biotechnology, Dallas, TX, USA) for immunohistochemical staining; the antibody clone used in the remaining two studies was not specified.

The table summarizes the key characteristics of the 14 studies included in the meta-analysis. It includes author name, publication year, country of origin, publication type, number of evaluated samples, specimen types analyzed, cutoff values used to define INSM1 positivity, and the antibody clone employed for immunohistochemical staining. CA, conference abstract; FA, full article; Surg, surgical specimen; PE, pleural effusion specimen; CSm, cytology smear; BBS, bronchoscopic biopsy specimen; Cyt, cytology specimen; ANS, any nuclear staining; NS, not specified. Santa Cruz = Santa Cruz Biotechnology, Dallas, TX, USA; catalog no. sc-271408. The H-score was calculated by multiplying the staining intensity (scored as 0–3) by the percentage of positive tumor cells, resulting in a score ranging from 0 to 300. For example, an H-score of 2 may correspond to weak intensity (score 1) observed in 2% of tumor cells (1 × 2 = 2).

Risk of bias and applicability were assessed using the QUADAS-2 tool (Figure 2). In the patient selection domain, three studies were rated as high risk of bias, primarily due to inappropriate case–control (two-gate) designs. The index test domain also showed high risk in three studies, mainly because the cut-off value for INSM1 positivity was determined post hoc, after reviewing the staining results, potentially introducing interpretation bias. The reference standard domain was judged to be unclear risk in most studies due to limited methodological description. In addition, the flow and timing domain was predominantly rated as low risk.

Regarding applicability concerns, most studies showed low concern. Some concern was noted in the patient selection domain, where three were rated as high concern and two as unclear. The index test and reference standard domains had generally low applicability concerns, suggesting that the populations, tests, and reference standards used in the included studies were appropriate for the review question.

(a) Three studies showed high risk in patent selection and index test domains. Reference standard risk was mostly unclear, while flow and timing was largely low risk. (b) Applicability concerns were generally low, with some concerns in patient selection. QUADAS-2, Quality Assessment of Diagnostic Accuracy Studies 2.

### 3.2. Diagnostic Accuracy of INSM-1 in NSCLC Model

The diagnostic performance of INSM1 for distinguishing LCNEC from other NSCLC was assessed across 11 studies (Figure 3). The pooled sensitivity was 0.67 (95% CI: 0.61–0.73), with strong heterogeneity (Chi-square = 43.17, *p* < 0.001; I^2^ = 76.8%). The pooled specificity was 0.97 (95% CI: 0.96–0.98), with moderate heterogeneity (Chi-square = 29.13, *p* = 0.001, I^2^ = 65.7%).

The SROC curve showed an AUC of 0.943, indicating very good diagnostic accuracy. The pooled diagnostic odds ratio (DOR) was 79.99 (95% CI: 38.37–166.75).

The pooled positive likelihood ratio (PLR) was 18.42 (95% CI: 10.08–33.67), and the pooled negative likelihood ratio (NLR) was 0.25 (95% CI: 0.14–0.43), both estimated using a random-effects model. These values indicate a strong ability of INSM1 to rule in and rule out LCNEC, respectively (Appendix A).

The bivariate random-effects model produced results consistent with the primary analysis, with a pooled sensitivity of 0.742 (95% CI: 0.621–0.835), a specificity of 0.957 (95% CI: 0.927–0.975), and an AUC of 0.943, indicating good robustness of our findings.

No significant threshold effect was observed. The Spearman correlation coefficient between sensitivity and 1 − specificity was −0.35 (*p* = 0.29), which did not meet the predefined criteria (r ≥ 0.60 and *p* < 0.05).

These findings indicate that INSM1 demonstrates excellent specificity but fair sensitivity for differentiating LCNEC from other NSCLC, supporting its clinical utility as a confirmatory marker in this diagnostic setting. The observed I^2^ of 76.8% for sensitivity suggests strong heterogeneity, indicating that variability among the included studies was likely due to real differences in study populations, staining protocols, or cutoff criteria rather than chance alone. The I^2^ of 65.7% for specificity reflects moderate heterogeneity, warranting cautious interpretation of the pooled estimate.

### 3.3. Diagnostic Accuracy of INSM-1 in Lung Cancer Model

The diagnostic accuracy of INSM1 for identifying NECs, including both SCLC and LCNEC, was evaluated across 14 studies (Figure 4). The pooled sensitivity was 0.86 (95% CI: 0.84–0.88), with strong heterogeneity (Chi-square = 74.26, *p* < 0.001; I^2^ = 82.5%). The pooled specificity was 0.97 (95% CI: 0.96–0.98), with moderate heterogeneity (Chi-square = 34.95, *p* = 0.001; I^2^ = 62.8%).

The SROC curve yielded an AUC of 0.974, indicating excellent diagnostic accuracy. The pooled diagnostic odds ratio (DOR) was 260.65 (95% CI: 122.36–555.24), representing a strong overall discriminatory ability of INSM1 for identifying pulmonary NEC.

The pooled positive likelihood ratio (PLR) was 25.40 (95% CI: 14.92–43.23), and the pooled negative likelihood ratio (NLR) was 0.10 (95% CI: 0.06–0.16), both estimated using a random-effects model. These metrics support INSM1′s high diagnostic utility for both confirming and excluding NECs (Appendix A).

The bivariate random-effects model produced a pooled sensitivity of 0.894 (95% CI: 0.848–0.927), a pooled specificity of 0.963 (95% CI: 0.942–0.977), and an AUC of 0.973. These findings were consistent with the primary analysis, supporting the robustness of the diagnostic performance estimates.

No significant threshold effect was observed. The Spearman correlation coefficient between sensitivity and 1 − specificity was −0.61 (*p* = 0.02), which did not meet the predefined criteria (r ≥ 0.60 and *p* < 0.05) for indicating a threshold effect.

These results demonstrate that INSM1 provides good sensitivity and excellent specificity for the diagnosis of pulmonary NECs, supporting its utility as a highly accurate immunohistochemical marker in this context. The strong heterogeneity in sensitivity (I^2^ = 82.5%) highlights substantial between-study variability, which may stem from differences in sample types, diagnostic thresholds, or inclusion criteria. The moderate heterogeneity in specificity (I^2^ = 62.8%) similarly suggests some variability, although the consistently high specificity across studies reinforces the robustness of this marker for ruling in NECs.

## 4. Discussion

To our knowledge, this is the first systematic review and meta-analysis to comprehensively assess the diagnostic utility of INSM1 in distinguishing SCLC and LCNEC from other subtypes of NSCLC. Given the clinical and pathological complexity of SCLC and LCNEC, accurate identification of neuroendocrine differentiation is essential for guiding treatment decisions and predicting prognosis [9,10,11]. While SCLC can often be diagnosed based on its characteristic morphological features, LCNEC requires confirmation of neuroendocrine differentiation through IHC [12,13]. However, morphology-based diagnosis can be challenging, especially in small or poorly preserved specimens, wherein immunostaining becomes critical [2].

Chromogranin A, synaptophysin, and CD56 are the most commonly used neuroendocrine markers and are officially recommended in current pathological practice [12]. Despite their widespread use, these markers exhibit notable limitations. Synaptophysin and CD56 are generally more sensitive but lack specificity, often staining a broad range of non-neuroendocrine tumors. Conversely, chromogranin A tends to be more specific but suffers from low sensitivity [14]. Although these markers are typically used in combination to improve diagnostic performance, nearly 10% of SCLCs may be negative or only focally positive for all three markers, and the sensitivity for LCNEC also remains suboptimal [12,16,17,19]. Moreover, as cytoplasmic markers, their interpretation can be problematic—particularly in cases of weak or focal staining—leading to potential confusion with nonspecific background signals [18].

Considering these limitations, INSM1 has recently emerged as a promising novel neuroendocrine marker [11,19]. INSM1 is a zinc-finger transcription factor involved in neuroendocrine differentiation during development [20]. Functionally, INSM1 exerts its regulatory effects by promoting the expression of classical neuroendocrine markers—chromogranin A, synaptophysin, and CD56—through activation of lineage-specific transcription factors such as ASCL1 and BRN2 [44]. It was originally cloned as IA-1 in 1992 by Goto et al. from a human insulinoma subtraction library and has since been found to be expressed in a variety of neuroendocrine neoplasms, including insulinomas, medullary thyroid carcinomas, pituitary adenomas, and pulmonary neuroendocrine neoplasms [20,24,25]. Notably, INSM1 has been demonstrated to be highly expressed in SCLC, carcinoid tumors, and in the SCLC-like molecular subtype of LCNEC, while generally absent in non-neuroendocrine NSCLCs [19,24].

Furthermore, INSM1′s nuclear staining pattern offers a distinct interpretive advantage over traditional cytoplasmic markers, particularly in small biopsy or cytology specimens, which are often the only available samples for diagnosis in advanced-stage lung cancer [18,45]. This combination of biological relevance, high specificity, and interpretive clarity underscores INSM1′s value in the diagnostic evaluation of pulmonary NECs.

Several studies have assessed the diagnostic performance of INSM1 in pulmonary NECs, reporting high sensitivity and specificity for distinguishing SCLC and LCNEC from other lung tumors [10,11,18,19,34,35,36,37,38,39,40,41,42,43]. However, reported estimates of sensitivity and specificity vary across studies, highlighting the need for a systematic review and meta-analysis to determine pooled performance metrics and clarify its diagnostic utility.

In a recent study, Sakakibara et al. reported a sensitivity of 92% and specificity of 94% for INSM1 in SCLC, along with a 68% positivity rate in LCNEC [11]. These findings are consistent with our meta-analysis, which demonstrated excellent specificity and reasonable sensitivity for both SCLC and LCNEC. While some individual studies reported lower sensitivity for LCNEC (as low as 42%), our analysis, based on 14 studies comprising 3,218 cases, provides a more statistically robust and comprehensive estimate. Notably, we observed substantial statistical heterogeneity across studies, particularly in pooled sensitivity estimates (I^2^ = 76.8% in the NSCLC model and 82.5% in the combined model). These high I^2^ values likely reflect both biological and methodological heterogeneity. Biologically, heterogeneity within LCNEC itself may have contributed to the observed variability. LCNEC can be stratified into molecular subtypes—SCLC-like and NSCLC-like—based on gene expression and mutational profiles [46]. INSM1 is more frequently expressed in SCLC-like LCNEC, whereas NSCLC-like subtypes may lack significant INSM1 expression, potentially lowering overall sensitivity in pooled analyses [19]. Methodologically, variability in study design, sample types, and INSM1 cutoff criteria may have contributed to the observed heterogeneity. However, no significant threshold effect was observed based on the Spearman correlation between sensitivity and 1 − specificity, suggesting that threshold variability may not have been a dominant source of heterogeneity. Although specificity estimates showed moderate heterogeneity (I^2^ = 65.7% in the NSCLC model and 62.8% in the combined model), their consistently high values reinforce the robustness of INSM1 as a confirmatory marker.

Our results are consistent with the findings of Staaf et al., who used an integrative analysis combining IHC and gene expression profiling in a systematic review of 39 prior studies [19]. They reported INSM1 positivity rates of 89% in SCLC (based on eight studies), 58% in LCNEC (four studies), and 1% in NSCLC (six studies). Their study provided valuable information by aggregating data across prior reports to estimate positivity rates. However, formal meta-analytic techniques were not employed. In our study, we conducted a quantitative synthesis pooling sensitivity, specificity, and SROC/AUC metrics, providing a complementary and statistically integrated estimate of INSM1′s diagnostic performance. We observed an AUC of 0.943 for distinguishing LCNEC from NSCLC and an AUC of 0.974 for differentiating combined SCLC and LCNEC from non-neuroendocrine tumors.

These results highlight the potential of INSM1 as a standalone neuroendocrine marker, particularly when traditional cytoplasmic markers yield equivocal or conflicting results. Its nuclear staining pattern, high specificity, and strong expression in high-grade neuroendocrine carcinomas render it especially valuable in the evaluation of small biopsy or cytology specimens, which are often the only materials available in advanced-stage disease. In resource-limited settings, the ability to rely on a single, robust, and interpretable marker could streamline diagnostic workflows and improve accuracy.

INSM1 has been proposed not only as a useful adjunct but also as a potential first-line marker for confirming neuroendocrine differentiation in pulmonary neuroendocrine neoplasms, particularly SCLC and LCNEC [11,14]. Our findings reinforce this position, demonstrating excellent specificity and very good-to-excellent overall diagnostic accuracy for identifying high-grade neuroendocrine carcinomas.

Clinically, this distinction is of great importance. SCLC and LCNEC are aggressive malignancies characterized by rapid proliferation, early metastasis, and poor prognosis and are typically managed with systemic chemotherapy [2,3,4,5,6]. Accurate pathological classification is critical for initiating appropriate treatment and predicting prognosis. Therefore, a reliable and efficient marker such as INSM1 could play a central role in both routine diagnostic practice and clinical research.

Despite the promising findings, this study has several limitations. First, although most included studies attempted to avoid overrepresentation by excluding duplicate samples from the same patient, one study reported two SCLC samples from a single patient that could not be separated, potentially introducing minor sampling bias. Second, definitions of INSM1 positivity varied across studies, with some accepting weakly positive staining or applying low cutoff thresholds (e.g., 0–5% positivity). While these thresholds may reflect real-world diagnostic practice, they limit cross-study comparability, and we were unable to propose a unified cutoff. Third, incorporation bias may be present in studies where pathologists were not clearly blinded to the index test (i.e., INSM1 expression) when making the reference standard diagnosis. Fourth, we did not perform a formal assessment of publication bias, as current methods such as Deeks’ funnel plot have limited reliability in the presence of heterogeneity [47]. Furthermore, no validated methods currently exist for publication bias assessment in bivariate meta-analysis models [47]. This decision is consistent with recommendations from the Cochrane Handbook and prior methodological studies [47,48].

Future studies should aim to establish standardized criteria for INSM1 immunohistochemical interpretation, including optimal cutoff thresholds and scoring systems, to promote inter-institutional consistency. Further validation in small biopsy and cytology specimens is also necessary, as SCLC is often diagnosed at advanced stages with limited tissue availability.

## 5. Conclusions

This meta-analysis demonstrates that INSM1 exhibits excellent specificity and very-good-to-excellent overall diagnostic performance for identifying pulmonary neuroendocrine carcinomas, specifically SCLC and LCNEC. Our findings support the integration of INSM1 into diagnostic workflows, especially when conventional markers yield ambiguous results or are technically limited. Given its nuclear localization, interpretive clarity, and superior diagnostic performance, INSM1 represents a valuable addition to the diagnostic armamentarium in thoracic pathology.

## Figures and Tables

**Figure 1 cancers-17-02544-f001:**
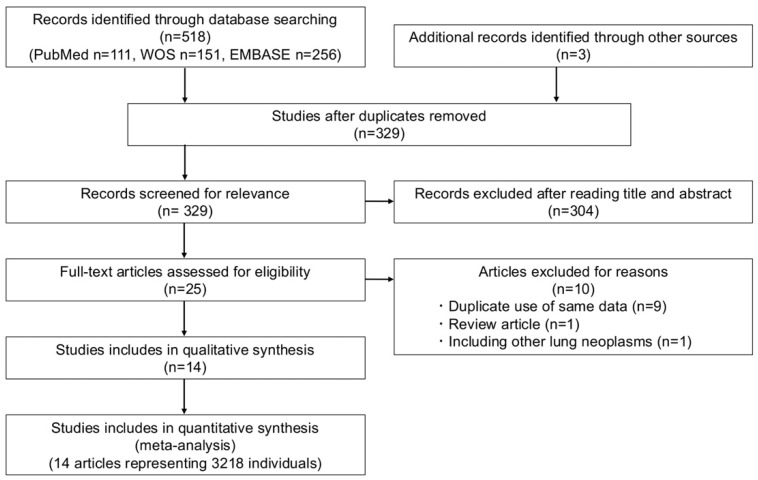
PRISMA flow diagram.

**Figure 2 cancers-17-02544-f002:**
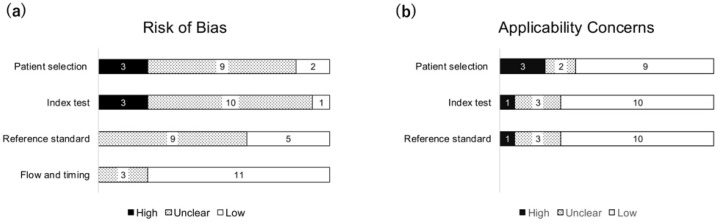
Risk of bias and applicability concerns assessed using the QUADAS-2 tool. (**a**) Risk of bias, and (**b**) Applicability concerns.

**Figure 3 cancers-17-02544-f003:**
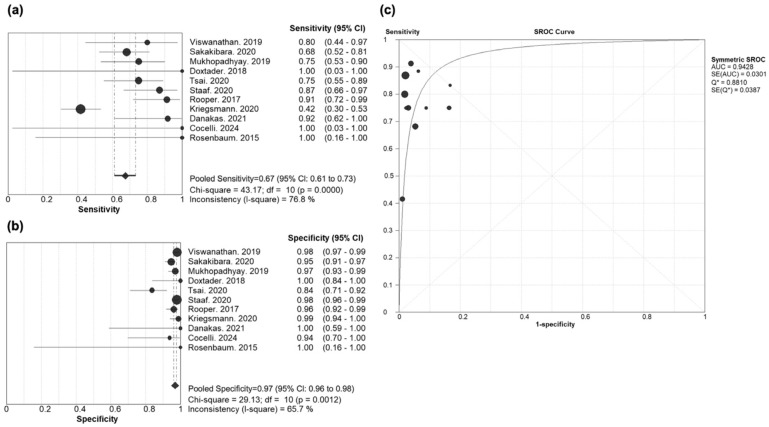
Diagnostic test accuracy of INSM1 in the NSCLC model. (**a**) Sensitivity, (**b**) specificity, and (**c**) SROC curve. (**a**) Each dot represents the point estimate of sensitivity for an individual study; the size of the dot reflects the weight of the study in the pooled analysis [7,10,11,18,19,34,36,37,41,42,43]. Horizontal lines indicate the 95% confidence interval (CI) for each study. Diamonds represent the pooled estimates, with pooled sensitivity of 0.67 (95% CI: 0.61–0.73). (**b**) Each dot represents the point estimate of specificity for an individual study; the size of the dot reflects the weight of the study in the pooled analysis [7,10,11,18,19,34,36,37,41,42,43]. Horizontal lines indicate the 95% confidence interval (CI) for each study. Diamonds represent the pooled estimates, with pooled specificity of 0.97 (95% CI: 0.96–0.98). (**c**) Each dot represents an individual study plotted in ROC space (sensitivity vs. 1 − specificity) [7,10,11,18,19,34,36,37,41,42,43]. The AUC of the SROC curve was 0.943, indicating very good diagnostic accuracy with excellent specificity and fair sensitivity. AUC, area under the curve; SE, standard error; SROC curve, summary receiver operating characteristic curve; ROC, receiver operating characteristic; INSM1, insulinoma-associated protein 1; NSCLC, non–small cell lung cancer.

**Figure 4 cancers-17-02544-f004:**
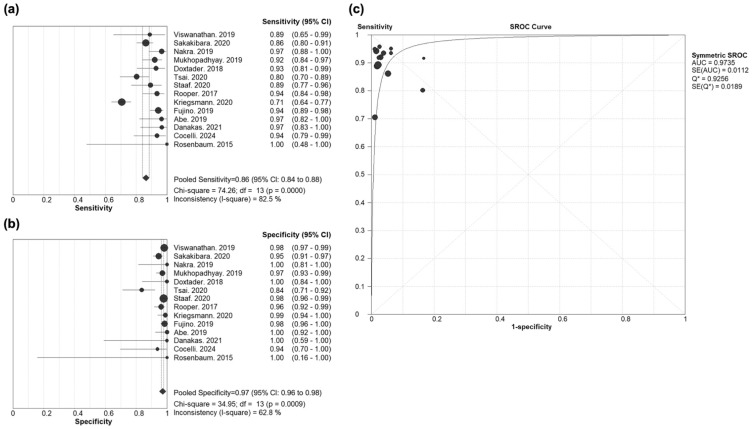
Diagnostic test accuracy of INSM1 in the lung cancer model. (**a**) Sensitivity, (**b**) specificity, and (**c**) SROC curve. (**a**) Each dot represents the point estimate of sensitivity for an individual study; the size of the dot reflects the weight of the study in the pooled analysis [7,10,11,18,19,34,35,36,37,39,40,41,42,43]. Horizontal lines indicate the 95% confidence interval (CI) for each study. Diamonds represent the pooled estimates, with pooled sensitivity of 0.86 (95% CI: 0.84–0.88). (**b**) Each dot represents the point estimate of sensitivity/specificity for an individual study; the size of the dot reflects the weight of the study in the pooled analysis [7,10,11,18,19,34,35,36,37,39,40,41,42,43]. Horizontal lines indicate the 95% confidence interval (CI) for each study. Diamonds represent the pooled estimates, with pooled specificity of 0.97 (95% CI: 0.96–0.98). (**c**) Each dot represents an individual study plotted in ROC space (sensitivity vs. 1 − specificity) [7,10,11,18,19,34,35,36,37,39,40,41,42,43]. The AUC of the SROC curve was 0.974, demonstrating excellent diagnostic accuracy for INSM1 in detecting NECs. AUC, area under the curve; SE, standard error; SROC curve, summary receiver operating characteristic curve; ROC, receiver operating characteristic; INSM1, insulinoma-associated protein 1; NECs, neuroendocrine carcinomas.

**Table 2 cancers-17-02544-t002:** Characteristics of included studies.

Author	Year	Country	Publication Types	No. of All Samples	Specimen Types	Cutoff Value	Antibody
Viswanathan	2019	USA	FA	508	Surg/PE	PE: ≥2+, Surg: ≥1+ in >5%	Santa Cruz
Sakakibara	2020	Japan	FA	427	Surg/BBS	H-score ≥ 5	Santa Cruz
Nakra	2019	India	FA	77	Csm/BBS	≥weak	Santa Cruz
Mukhopadhyay	2019	USA	FA	255	Surg/BBS	≥ANS	Santa Cruz
Doxtader	2018	USA	FA	63	PE/Surg/BBS	≥ANS	Santa Cruz
Tsai	2020	USA	FA	131	Surg/BBS	H-score of ≥50	Santa Cruz
Staaf	2020	Sweden	FA	664	Surg/BBS	≥Weak staining in >0%	Santa Cruz
Rooper	2017	USA	FA	218	BBS	≥1+	Santa Cruz
Kriegsmann	2020	Germany	FA	312	Surg	≥weak staining in ≥1%	Santa Cruz
Fujino	2019	Japan	CA	396	Surg/Cyt	NS	NS
Abe	2019	Japan	FA	76	PE	≥8.68%	Santa Cruz
Danakas	2021	USA	CA	37	BBS/Surg	NS	Santa Cruz
Cocelli	2024	USA	CA	47	Cyt	NS	NS
Rosenbaum	2015	USA	FA	7	NS	≥2%	Santa Cruz

## Data Availability

No new datasets were generated or analyzed in this study. All data analyzed were extracted from previously published studies, which are cited in the reference list.

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
