# Peer review of "Diagnostic Accuracy of Insulinoma-Associated Protein 1 in Pulmonary Neuroendocrine Carcinomas: A Systematic Review and Meta-Analysis"

_cancers, 2025, doi:10.3390/cancers17152544_

Round 1
Reviewer 1 Report
Comments and Suggestions for Authors
The manuscript titled “Diagnostic accuracy of Insulinoma-associated protein 1 in pulmonary neuroendocrine carcinomas: a systematic review and meta-analysis” addresses an interesting and clinically relevant question. The role of INSM1 in pulmonary neuroendocrine tumors (NETs) is still evolving, and a systematic synthesis of current evidence is welcome.
However, the study presents several major issues that must be addressed before it can be considered for publication:
-
Methodological Weaknesses:
-
The meta-analysis lacks a clear protocol registration (e.g., PROSPERO), which is standard for systematic reviews and meta-analyses.
-
No formal risk of bias assessment was performed on the included diagnostic studies (e.g., QUADAS-2), which weakens the reliability of the conclusions.
-
Heterogeneity is insufficiently explored. The I² values are mentioned but not interpreted. No subgroup analysis or meta-regression is provided to account for possible sources of heterogeneity (e.g., tumor subtype, antibody clone, study setting).
-
-
Study Selection and Data Extraction:
-
The inclusion criteria are overly vague. The authors should clearly distinguish between high-grade (e.g., LCNEC, SCLC) and low-grade (e.g., carcinoid) tumors.
-
The data presented for the diagnostic performance of INSM1 are not consistently aligned with clinical practice: was the index test used alone or compared to standard markers such as Chromogranin A, Synaptophysin, or CD56?
-
-
Statistical Analysis and Presentation:
-
The forest plots are too basic, and publication bias assessment is missing.
-
Confidence intervals for pooled sensitivity and specificity should be emphasized in the results and figures.
-
No mention of bivariate random-effects models or HSROC curves, which are standard in diagnostic accuracy meta-analyses.
-
-
Terminology and Clarity:
-
The manuscript should better differentiate between “pulmonary neuroendocrine tumors” and “pulmonary neuroendocrine carcinomas.” These are not synonyms: the former includes both typical/atypical carcinoids (low- and intermediate-grade) and high-grade carcinomas (LCNEC/SCLC).
-
The term “diagnostic accuracy” is used without specifying what thresholds or gold standards were used.
-
While the language is generally understandable, multiple grammatical errors and awkward expressions are present. A professional English editing service is strongly recommended to improve fluency and clarity.
Author Response
Reply to the Review Report (Reviewer 1)
We thank the reviewer for their thoughtful and constructive feedback on our manuscript entitled “Diagnostic accuracy of Insulinoma-associated protein 1 in pulmonary neuroendocrine carcinomas: a systematic review and meta-analysis.” We appreciate the reviewer’s recognition of the clinical relevance of our research and the value of systematically synthesizing current evidence on INSM1.
We fully acknowledge the importance of the major concerns raised and have carefully revised the manuscript to address each of these points. Our detailed responses are provided below. We believe that these revisions have significantly improved the clarity, accuracy, and overall quality of the manuscript.
Comments 1.1:
Methodological Weaknesses:
The meta-analysis lacks a clear protocol registration (e.g., PROSPERO), which is standard for systematic reviews and meta-analyses.
Response 1.1:
We respectfully disagree. The paper does have a clear statement of protocol registration in the Study Overview subsection, in the first paragraph of Methods section (page 3). The protocol of this study was registered on the University Hospital Medical Information Network Clinical Trials Registry (UMIN000056564). Like PROSPERO, UMIN is an ICMJE (International Committee of Medical Journal Editors) certified official protocol registry and is therefore acceptable.
Comment 1.2:
No formal risk of bias assessment was performed on the included diagnostic studies (e.g., QUADAS-2), which weakens the reliability of the conclusions.
Response 1.2:
We respectfully disagree, as we did perform a formal risk of bias assessment, as noted in the Risk of Bias subsection, in the seventh paragraph of Methods section (page 4). We used the QUADAS-2 tool to assess the quality of the included diagnostic studies. The results of the QUADAS-2 assessment were used for descriptive purposes only and did not influence study inclusion, statistical weighting, or exclusion in the meta-analysis. However, we did take the results of the quality assessment into account when interpreting our findings. Specifically, in the penultimate paragraph of the Discussion section, we addressed the potential for incorporation bias—our third noted limitation—arising in studies where pathologists may not have been clearly blinded to the index test (i.e., INSM1 expression) during reference standard diagnosis.
Comment 1.3:
Heterogeneity is insufficiently explored. The I² values are mentioned but not interpreted. No subgroup analysis or meta-regression is provided to account for possible sources of heterogeneity (e.g., tumor subtype, antibody clone, study setting).
Response 1.3:
We agree that exploring heterogeneity is an important aspect of this meta-analysis. The reviewed paper addressed this issue in the second half of the sixth paragraph of the Discussion section (page 12). We believe the substantial heterogeneity in pooled sensitivities is attributable to both biological heterogeneity (e.g., differences in tumor subtypes such as SCLC-like vs. NSCLC-like LCNEC) and methodological heterogeneity (e.g., variations in study design, staining protocols, and cutoff criteria). Although variability in cutoff criteria could potentially contribute to threshold effects, we confirmed that no significant threshold effect was present based on the Spearman correlation between sensitivity and 1 – specificity, as mentioned in the third paragraph of the Statistical Analysis subsection within the Methods section (page 5), and in the fifth paragraph of the Diagnostic Accuracy of INSM1 in the NSCLC Model subsection and the fifth paragraph of the Diagnostic Accuracy of INSM1 in the Lung Cancer Model subsection within the Results section.
Regarding subgroup analysis and meta-regression, we acknowledge their utility in investigating heterogeneity. However, due to the limited number of studies included in our analysis, we believe such analyses would be underpowered and potentially unreliable. Therefore, we decided not to perform subgroup analysis or meta-regression in this study.
Comments 1.4:
Study Selection and Data Extraction:
The inclusion criteria are overly vague. The authors should clearly distinguish between high-grade (e.g., LCNEC, SCLC) and low-grade (e.g., carcinoid) tumors.
Response 1.4:
We appreciate the reviewer’s insightful comment. We have accordingly revised the Study Selection subsection (page 3-4) to clarify that our analysis focused exclusively on high-grade neuroendocrine carcinomas (i.e., SCLC and LCNEC)
Comment 1.5:
The data presented for the diagnostic performance of INSM1 are not consistently aligned with clinical practice: was the index test used alone or compared to standard markers such as Chromogranin A, Synaptophysin, or CD56?
Response 1.5:
Thank you for your insightful comment. We have clarified this point in the Study Selection subsection (page 3-4). Studies were included regardless of whether INSM1 was evaluated alone or in comparison with conventional neuroendocrine markers such as Chromogranin A, Synaptophysin, or CD56. However, it is worth noting that the majority of the included studies did compare the diagnostic performance of INSM1 with one or more of these standard markers. For the purpose of this meta-analysis, we focused exclusively on extracting and analyzing the diagnostic performance of INSM1.
Comments 1.6:
Statistical Analysis and Presentation:
The forest plots are too basic, and publication bias assessment is missing.
Response 1.6:
Thank you for your comment regarding the forest plots.
While we understand the reviewer’s concern, we would like to note that the current format of the forest plots follows the standard and widely accepted structure for diagnostic test accuracy meta-analyses. Forest plots in their basic form are commonly used in systematic reviews because they present key quantitative information (e.g., sensitivity, specificity, confidence intervals) in a clear and easily interpretable manner. We believe that retaining the standard format improves readability and accessibility for a broad range of readers, including clinicians and non-statistical audiences. For these reasons, and in line with common practice in DTA reviews, we have maintained the current forest plot structure.
Also, as for publication bias, it is well recognized that assessing publication bias in diagnostic test accuracy (DTA) meta-analyses is more challenging than in intervention reviews due to several methodological complexities. These include typically high diagnostic odds ratios (DORs), variation in positivity thresholds across studies, and imbalanced numbers of diseased and non-diseased participants—all of which affect precision and funnel plot symmetry, as discussed by van Enstet al., Investigation of publication bias in meta-analyses of diagnostic test accuracy: a meta-epidemiological study (BMC Med Res Methodol. 2014;14:70).
While Deeks’ test is recommended for assessing publication bias in DTA meta-analyses, it has limited power in the presence of heterogeneity—an inherent characteristic of DTA studies—which limits its interpretability.
Based on the above considerations, we did not perform publication bias analysis in our bivariate DTA meta-analysis. We have also added statements explaining this decision and the rationale behind it as the fourth limitation of this study, in the penultimate paragraph of the Discussion section (page 13).
Comment 1.7:
Confidence intervals for pooled sensitivity and specificity should be emphasized in the results and figures.
Response 1.7:
We thank the reviewer for this important comment. In response, we have added the 95% confidence intervals for pooled sensitivity and specificity to the Abstract. Additionally, we modified the figure captions to explicitly state “95% CI” (e.g., 0.97 (95% CI: 0.96 to 0.98)) for clarity (Figure 3, Figure 4).
Finally, we have enhanced the visual emphasis of the confidence intervals in the forest plots by thickening the corresponding lines (Figure 3, Figure 4).
We hope these changes improve the readability and interpretation of the results as suggested.
Comment 1.8:
No mention of bivariate random-effects models or HSROC curves, which are standard in diagnostic accuracy meta-analyses.
Response 1.8:
Thank you for your valuable suggestion. As suggested, the bivariate random-effects model analysis was conducted and has been added to the second paragraph of the Statistical Analysis subsection in the Methods section (page 5), and to the fourth paragraph of the Diagnostic Accuracy of INSM1 in the NSCLC Model and the Diagnostic Accuracy of INSM1 in the Lung Cancer Model subsections in the Results section (page 8-9). This sensitivity analysis supports the robustness of our primary findings.
Comments 1.9:
Terminology and Clarity:
The manuscript should better differentiate between “pulmonary neuroendocrine tumors” and “pulmonary neuroendocrine carcinomas.” These are not synonyms: the former includes both typical/atypical carcinoids (low- and intermediate-grade) and high-grade carcinomas (LCNEC/SCLC).
Response 1.9:
Thank you for your insightful comment. We fully agree that “pulmonary neuroendocrine tumors” and “pulmonary neuroendocrine carcinomas” are not interchangeable terms. In response, we have carefully revised the manuscript to consistently distinguish between these entities. Specifically, we now use “pulmonary neuroendocrine neoplasms” to refer to the full spectrum, including typical and atypical carcinoids, and reserve “pulmonary neuroendocrine carcinomas” (NECs) for LCNEC and SCLC. These corrections have been implemented throughout the manuscript to improve clarity and accuracy (e.g., in the final paragraph of the Introduction section and other relevant parts).
We also clarified that the focus of our study is specifically on pulmonary neuroendocrine carcinomas—namely LCNEC and SCLC—as opposed to low- and intermediate-grade pulmonary neuroendocrine tumors, in the first part of the Study Selection subsection (page 3).
Comment 1.10:
The term “diagnostic accuracy” is used without specifying what thresholds or gold standards were used.
Responce 1.10:
Thank you for your comment.
The gold standard for diagnosis was defined as the final pathological diagnosis by experienced pathologists, as described in the final paragraph of the Study Selection subsection in the Methods section (page 4). We have added a clarifying sentence to make this more explicit and easier to locate in the same paragraph.
We agree that thresholds are an important factor in evaluating diagnostic accuracy. As cutoff values varied among the included studies and were inconsistently reported, we were unable to calculate a pooled threshold, which is presented as the second limitation of this study in the penultimate paragraph of the Discussion section (page 13). To address this, we have clarified in the Outcomes subsection of the Methods section (page 3-4) that all INSM1-positive cases—as defined by the original studies (regardless of the cutoff criteria)—were treated as positive in our analysis.
Comment 1.11
While the language is generally understandable, multiple grammatical errors and awkward expressions are present. A professional English editing service is strongly recommended to improve fluency and clarity.
Response 1.11
Thank you for your suggestion. In response, we have had the paper edited for fluency and clarity.
Reviewer 2 Report
Comments and Suggestions for Authors
The authors have adopted a unique approach to validating the INSM1 marker using meta-analysis. I found the manuscript interesting; however, I suggest the authors address the following comments, which I believe will strengthen the overall merit of the paper.
1. Please state that threshold effects were evaluated by plotting individual studies in ROC space and calculating the Spearman correlation between logit-sensitivity and logit(1 − specificity), using |r| ≥ 0.60 with P < 0.05 as the decision rule.
2. Calculate and report the positive likelihood ratio (PLR), negative likelihood ratio (NLR), and diagnostic odds ratio (DOR) using either random- or fixed-effects models, and confirm that each estimate is accompanied by a 95% confidence interval.
3. Assess publication bias using Deeks’ funnel-plot asymmetry test (e.g., in Stata using midast_das) and consider P < 0.10 as indicative of potential bias.
4. Clarify how studies assessed with QUADAS-2 were handled in the analysis, particularly if any domains of high risk or applicability concerns influenced inclusion, weighting, or interpretation of results.
Author Response
Reply to the Review Report (Reviewer 2)
We sincerely thank the reviewer for their positive feedback and for recognizing the uniqueness of our meta-analytic approach to validating INSM1. We appreciate your thoughtful comments and suggestions, which we believe have helped improve the clarity and overall quality of the manuscript. We have addressed each point in detail below.
Comments 2.1:
Please state that threshold effects were evaluated by plotting individual studies in ROC space and calculating the Spearman correlation between logit-sensitivity and logit(1 − specificity), using |r| ≥ 0.60 with P < 0.05 as the decision rule.
Response 2.1:
Thank you for your thoughtful comment. As suggested, we evaluated potential threshold effects by calculating the Spearman correlation coefficient between sensitivity and 1 − specificity across studies. Although the correlation coefficient exceeded the predefined cutoff of 0.60 in absolute value (r = –0.61, p = 0.02), the correlation was negative, whereas a true threshold effect is typically characterized by a positive correlation between sensitivity and 1 − specificity, or equivalently, a negative correlation between sensitivity and specificity (J Rheum Dis. 2018;25(1):3–10). Therefore, this result does not support the presence of a threshold effect.
We also note that Spearman correlation is a rank-based non-parametric measure, and thus applying logit transformation to sensitivity and 1 − specificity does not affect the ranks or the correlation result. For this reason, we used the original values for the calculation. This explanation has been added to the third paragraph of the Statistical Analysis subsection in the Methods section (page 5) and to the fifth paragraph of both the Diagnostic Accuracy of INSM1 in the NSCLC Model (page 8) and the Diagnostic Accuracy of INSM1 in the Lung Cancer Model (page 10) subsections in the Results section.
Comments 2.2:
Calculate and report the positive likelihood ratio (PLR), negative likelihood ratio (NLR), and diagnostic odds ratio (DOR) using either random- or fixed-effects models, and confirm that each estimate is accompanied by a 95% confidence interval.
Response 2.2:
Thank you for your valuable comment. As recommended, we have calculated and reported the pooled positive likelihood ratio (PLR), negative likelihood ratio (NLR), and diagnostic odds ratio (DOR), each accompanied by a 95% confidence interval.
All estimates were derived using a random-effects model in Meta-DiSc version 1.4, and the results have been added to the third paragraph of both the Diagnostic Accuracy of INSM1 in the NSCLC Model (page 8) and the Diagnostic Accuracy of INSM1 in the Lung Cancer Model (page 9) subsections in the Results section. These additions provide a more comprehensive summary of the diagnostic performance of INSM1.
We believe that these metrics (PLR, NLR, and DOR) provide clinically meaningful estimates of diagnostic utility beyond sensitivity and specificity alone. Likelihood ratios facilitate post-test probability estimation in clinical decision-making, and the DOR offers a single, unified measure of overall test performance. Inclusion of these values enhances the interpretability and clinical relevance of our findings.
Comments 2.3:
Assess publication bias using Deeks’ funnel-plot asymmetry test (e.g., in Stata using midast_das) and consider P < 0.10 as indicative of potential bias.
Response 2.3:
Thank you very much for your thoughtful recommendation. It is well recognized that assessing publication bias in diagnostic test accuracy (DTA) meta-analyses is more challenging than in intervention reviews due to several methodological complexities. These include typically high diagnostic odds ratios (DORs), variation in positivity thresholds across studies, and imbalanced numbers of diseased and non-diseased participants—all of which affect precision and funnel plot symmetry, as discussed by van Enstet al., Investigation of publication bias in meta-analyses of diagnostic test accuracy: a meta-epidemiological study (BMC Med Res Methodol. 2014;14:70).
As the reviewer suggests, while Deeks’ test is recommended, it has limited power in the presence of heterogeneity—an inherent characteristic of DTA studies—and therefore its interpretation is limited. For these reasons, we did not perform this analysis.
We have also added statements explaining this decision and the rationale behind it as the fourth limitation of this study, in the penultimate paragraph of the Discussion section (page 13).
Comments 2.4:
Clarify how studies assessed with QUADAS-2 were handled in the analysis, particularly if any domains of high risk or applicability concerns influenced inclusion, weighting, or interpretation of results.
Response 2.4:
Thank you for your comment. The results of the QUADAS-2 assessment were used for descriptive purposes only and did not influence study inclusion, statistical weighting, or exclusion in the meta-analysis. We also did not conduct subgroup analyses based on QUADAS-2 domains. However, we did take the results of the quality assessment into account when interpreting our findings. Specifically, in the penultimate paragraph of the Discussion section (page 13), we addressed the potential for incorporation bias—our third noted limitation—arising in studies where pathologists may not have been clearly blinded to the index test (i.e., INSM1 expression) during reference standard diagnosis.
Additionally, we added a paragraph to the Risk of Bias subsection of the Methods section (page 4) to clarify how the QUADAS-2 results were used in our analysis.
Reviewer 3 Report
Comments and Suggestions for Authors
The authors of the systematic review entitled “Diagnostic accuracy of Insulinoma-associated protein 1 in pulmonary neuroendocrine carcinomas: a systematic review and meta-analysis” present a manuscript detailing a meta-analysis of 14 studies investigating INSM1 in two lung cancer subtypes, LCNEC and SCLC, in order to assess its diagnostic potential.
The authors should address the following issues:
Across the manuscript, fix the citations position (they should appear before the dot) from e.g., “prognostication. 7-9” to “prognostication 7-9.”
Please check journal policies regarding abbreviations position within the manuscript.
Please check the journal's policies on footers when using tables, as they appear to be incorrectly placed throughout the manuscript.
Introduction
Page 2. Include information regarding the incidence of both LCNEC and SCLC lung cancer subtypes in the population.
Page 2-3. Its good practice to link biomarkers to a database accession to future-proof name changes, e.g., Synaptophysin (P08247) from UniProtKB/ Swiss-Prot https://www.uniprot.org/uniprotkb/P08247.
Page 3. Add range of diagnostic accuracy for INSM1 (Recent studies have suggested high diagnostic accuracy…[...]).
Methods
Study Search
Page 3. Fix wrong numbering of Supplementary table: Supplementary Table 1. Search strategy for systematic review, while in-text is pointing to table S2 “are provided as supplementary material. (Table S2).”
Page 4. I would add a citation to QUADAS-2 on the first instance it occurs, i.e., “Quality Assessment of Diagnostic Accuracy Studies-2 (QUADAS-2) evaluation.”
Use of Generative AI Tools
Page 4. Missing figure number in the sentence, i.e., “The figure outlines the two diagnostic models assessed in this study.”
Outcomes
Redo this section altogether with table 1, since it is highly confusing.
Results
Page 5. Misspelled results: “RESLUTS”
Page 6. Check journal policies regarding abbreviations position in the manuscript, but shouldn’t go in this section, “All abbreviations are defined as follows: INSM1, insulinoma-associated protein 1; LCNEC, large cell neuroendocrine carcinoma; WOS, Web of Science.”
Page 6, table 2. Spell out the meaning of “ANS”, “CB”, “SG”, and all other abbreviations used.
Page 8-9, figure 3-4. Improve figure caption by describing the meaning of elements of the forest plot such as: dot size meaning; shapes (circle, diamond), line length for each study.
Page 8-9, figure 3-4. Generate confidence bands at 95% for the SROC curves (C).
Discussion
Include the absence of an assessment for publication bias as a limitation of this meta-analysis.
Author Response
Reply to the Review Report (Reviewer 3)
We would like to thank the reviewer for their careful reading of our manuscript entitled “Diagnostic accuracy of Insulinoma-associated protein 1 in pulmonary neuroendocrine carcinomas: a systematic review and meta-analysis.” We sincerely appreciate the constructive comments and suggestions. Below, we provide detailed responses to each of the points raised. We believe that the revisions made in response to these comments have strengthened the quality and clarity of our work.
Comments 3.1:
Across the manuscript, fix the citations position (they should appear before the dot) from e.g., “prognostication. 7-9” to “prognostication 7-9.”
Response 3.2:
Thank you for your comment. As suggested, we have reviewed the entire manuscript and corrected the placement of all in-text citations to ensure they appear before the period, in accordance with journal formatting guidelines.
Comments 3.2:
Please check journal policies regarding abbreviations position within the manuscript.
Response 3.2:
Thank you for your helpful comment. We identified several abbreviations used in the figures and tables that were not defined in the abbreviation section. We have corrected this by ensuring that all abbreviations are now defined.
Additionally, we moved the definitions immediately after the figure and table legends to clarify that the abbreviations are specific to the content within each figure or table.
Comments 3.3:
Please check the journal's policies on footers when using tables, as they appear to be incorrectly placed throughout the manuscript.
Response 3.2:
Thank you for your comment regarding the placement of table footers.
We have carefully reviewed the journal’s submission guidelines but could not locate any specific instructions regarding the placement of table footers or legends. In accordance with standard academic formatting practices, we have positioned the table legends immediately below each table and the figure legends directly beneath each figure. We hope this placement is acceptable, but we are happy to revise it further should the editorial team have specific formatting preferences.
Comments 3.4:
Introduction
Page 2. Include information regarding the incidence of both LCNEC and SCLC lung cancer subtypes in the population.
Response 3.4:
Thank you for your comment. As suggested, we have included epidemiological information regarding the incidence of pulmonary large cell neuroendocrine carcinoma (LCNEC) and small cell lung cancer (SCLC) in the revised manuscript. Specifically, we now state that LCNEC and SCLC comprise approximately 2–3% and 15% of all lung cancers, respectively. This information has been added to the second paragraph of Introduction section (page 2).
Comments 3.5:
Page 2-3. Its good practice to link biomarkers to a database accession to future-proof name changes, e.g., Synaptophysin (P08247) from UniProtKB/ Swiss-Prot https://www.uniprot.org/uniprotkb/P08247.
Responce 3.5:
Thank you for your suggestion. In accordance with your recommendation, we have added UniProtKB/Swiss-Prot accession numbers for the relevant biomarkers (e.g., Synaptophysin [P08247], Chromogranin A [P10645], CD56/NCAM1 [P13591], and INSM1 [Q01101]) to the fourth paragraph of the Introduction section (page 2) to ensure clarity and future-proof referencing.
Comments 3.6:
Page 3. Add range of diagnostic accuracy for INSM1 (Recent studies have suggested high diagnostic accuracy…[...]).
Response 3.6:
Thank you for your comment. As suggested, we have added the reported range of diagnostic accuracy for INSM1 to the penultimate paragraph of the Discussion section (page 3). Specifically, we now state that recent studies have suggested high diagnostic accuracy of INSM1 in identifying pulmonary neuroendocrine tumors, particularly SCLC and LCNEC, with reported sensitivity ranging from 68% to 95% and specificity from 95% to 99%.
Comments 3.7:
Methods
Study Search
Page 3. Fix wrong numbering of Supplementary table: Supplementary Table 1. Search strategy for systematic review, while in-text is pointing to table S2 “are provided as supplementary material. (Table S2).”
Response 3.7:
Thank you for pointing this out. We have corrected the mismatch between the in-text citation and the Supplementary Table numbering. The reference to " Supplementary Table 1" in the main text now correctly reads " Supplementary Table 2" to match the actual table in the supplementary (page 3).
Comments 3.8:
Page 4. I would add a citation to QUADAS-2 on the first instance it occurs, i.e., “Quality Assessment of Diagnostic Accuracy Studies-2 (QUADAS-2) evaluation.”
Response 3.8:
Thank you for your comment. As suggested, we have added the appropriate citation for QUADAS-2 at its first mention in the manuscript (i.e., “Quality Assessment of Diagnostic Accuracy Studies-2 [QUADAS-2] evaluation”) (page 4).
Comments 3.9:
Use of Generative AI Tools
Page 4. Missing figure number in the sentence, i.e., “The figure outlines the two diagnostic models assessed in this study.”
Response 3.9:
Thank you for your comment. We would like to clarify that the sentence “The figure outlines the two diagnostic models assessed in this study.” appears as part of the figure caption, not in the main text. The corresponding models are already referenced in the main text as “Two analytical models were employed (Table 1):” in the first sentence of the Outcomes subsection (page 4).
However, we noted that Table 1 was previously inserted in a less appropriate location, so we have repositioned it immediately below the relevant paragraph to improve clarity and alignment with its first in-text citation.
Comments 3.10:
Outcomes
Redo this section altogether with table 1, since it is highly confusing.
Response 3.10:
Thank you for your comment. In response, we have fully revised the Outcomes subsection and Table 1 (page 4-5) to improve clarity. To enhance visual distinction, we applied shading to the table: the positive group (LCNEC and SCLC) is now shown in dark gray, while the negative group (NSCLC other than LCNEC) is shown in light gray.
We also recognize that the relationship between LCNEC and NSCLC may have been unclear. To address this, we have revised the figure to better illustrate that LCNEC is a subtype of NSCLC, but is classified as a neuroendocrine tumors that typically expresses INSM1, and was therefore grouped as INSM1-positive in our analysis.
Comments 3.11:
Results
Page 5. Misspelled results: “RESLUTS”
Response 3.11:
Thank you very much for pointing out. We corrected the spelling as suggested(page 6).
Comments 3.12:
Page 6. Check journal policies regarding abbreviations position in the manuscript, but shouldn’t go in this section, “All abbreviations are defined as follows: INSM1, insulinoma-associated protein 1; LCNEC, large cell neuroendocrine carcinoma; WOS, Web of Science.”
Response 3.12:
Thank you for your comment. We would like to clarify that the list of abbreviations appears in the figure legend for Figure 1, not in the main text. To avoid confusion, we have ensured that all abbreviations used within each figure or table are defined immediately following the corresponding figure or table legend. We have removed any standalone abbreviation lists from inappropriate sections of the text and have placed all definitions in close proximity to where the abbreviations are actually used.
Comments 3.13:
Page 6, table 2. Spell out the meaning of “ANS”, “CB”, “SG”, and all other abbreviations used.
Response 3.13:
Thank you for your comment. We have addressed the abbreviation issues in Table 2 as follows:
“ANS” has been retained and is now defined as “any nuclear staining” in the abbreviation list.
“CB” (cell block) referred to pleural effusion specimens and has been replaced with the more descriptive term “pleural effusion” to improve clarity.
“SG” (surgical specimen) has been replaced with “surg”, which is used consistently throughout the table.
Comments 3.14:
Page 8-9, figure 3-4. Improve figure caption by describing the meaning of elements of the forest plot such as: dot size meaning; shapes (circle, diamond), line length for each study.
Response 3.14:
Thank you for your comment. To improve clarity, we have revised the figure captions for both Figure 3 and Figure 4 (page 9-10). Specifically, we added explanations for the elements of the forest plots, including the meaning of dot size, diamond shapes, and horizontal lines representing the 95% confidence intervals. For the SROC curves, we clarified that each dot represents an individual study plotted in ROC space (sensitivity vs. 1 − specificity).
Comments 3.15:
Page 8-9, figure 3-4. Generate confidence bands at 95% for the SROC curves (C).
Response 3.15:
Thank you for your comment. Unfortunately, our analysis software (Meta-DiSc1.4 and mada in R) does not support the generation of 95% confidence bands for SROC curves. While we acknowledge the value of displaying such intervals, we have instead provided pooled estimates with 95% confidence intervals for sensitivity, specificity, and diagnostic odds ratios, which reflect the uncertainty around diagnostic performance. We hope this provides sufficient statistical context for interpretation.
The DOR values were newly added in response to comments from another reviewer. Estimates of the diagnostic odds ratio (DOR) were derived using a random-effects model in Meta-DiSc version 1.4, and the results have been added to the third paragraph of both the Diagnostic Accuracy of INSM1 in the NSCLC Model (page 8) and the Diagnostic Accuracy of INSM1 in the Lung Cancer Model (page 9) subsections in the Results section.
Comments 3.16:
Discussion
Include the absence of an assessment for publication.
Response 3.16
Thank you very much for your thoughtful recommendation. It is well recognized that assessing publication bias in diagnostic test accuracy (DTA) meta-analyses is more challenging than in intervention reviews due to several methodological complexities. These include typically high diagnostic odds ratios (DORs), variation in positivity thresholds across studies, and imbalanced numbers of diseased and non-diseased participants—all of which affect precision and funnel plot symmetry, as discussed by van Enstet al., Investigation of publication bias in meta-analyses of diagnostic test accuracy: a meta-epidemiological study (BMC Med Res Methodol. 2014;14:70).
While Deeks’ test is recommended for assessing publication bias in DTA meta-analyses, it has limited power in the presence of heterogeneity—an inherent characteristic of DTA studies—and therefore its interpretation is limited.
Based on the above considerations, we did not perform publication bias analysis in our bivariate DTA meta-analysis. We have also added statements explaining this decision and the rationale behind it as the fourth limitation of this study, in the penultimate paragraph of the Discussion section (page 13).
We hope this clarifies our approach and enhances the transparency of our reporting.
Round 2
Reviewer 3 Report
Comments and Suggestions for Authors
Comments 3.1:
Across the manuscript, fix the citations position (they should appear before the dot) from e.g., “prognostication. 7-9” to “prognostication 7-9.”
Response 3.2:
Thank you for your comment. As suggested, we have reviewed the entire manuscript and corrected the placement of all in-text citations to ensure they appear before the period, in accordance with journal formatting guidelines.
[Rev] Corrected.
Comments 3.2:
Please check journal policies regarding abbreviations position within the manuscript.
Response 3.2:
Thank you for your helpful comment. We identified several abbreviations used in the figures and tables that were not defined in the abbreviation section. We have corrected this by ensuring that all abbreviations are now defined. Additionally, we moved the definitions immediately after the figure and table legends to clarify that the abbreviations are specific to the content within each figure or table.
[Rev] Corrected.
Comments 3.3:
Please check the journal's policies on footers when using tables, as they appear to be incorrectly placed throughout the manuscript.
Response 3.2:
Thank you for your comment regarding the placement of table footers.
We have carefully reviewed the journal’s submission guidelines but could not locate any specific instructions regarding the placement of table footers or legends. In accordance with standard academic formatting practices, we have positioned the table legends immediately below each table and the figure legends directly beneath each figure. We hope this placement is acceptable, but we are happy to revise it further should the editorial team have specific formatting preferences.
[Rev] Corrected.
Comments 3.4:
Introduction
Page 2. Include information regarding the incidence of both LCNEC and SCLC lung cancer subtypes in the population.
Response 3.4:
Thank you for your comment. As suggested, we have included epidemiological information regarding the incidence of pulmonary large cell neuroendocrine carcinoma (LCNEC) and small cell lung cancer (SCLC) in the revised manuscript. Specifically, we now state that LCNEC and SCLC comprise approximately 2–3% and 15% of all lung cancers, respectively. This information has been added to the second paragraph of Introduction section (page 2).
[Rev] Corrected.
Comments 3.5:
Page 2-3. Its good practice to link biomarkers to a database accession to future-proof name changes, e.g., Synaptophysin (P08247) from UniProtKB/ Swiss-Prot https://www.uniprot.org/uniprotkb/P08247.
Responce 3.5:
Thank you for your suggestion. In accordance with your recommendation, we have added UniProtKB/Swiss-Prot accession numbers for the relevant biomarkers (e.g., Synaptophysin [P08247], Chromogranin A [P10645], CD56/NCAM1 [P13591], and INSM1 [Q01101]) to the fourth paragraph of the Introduction section (page 2) to ensure clarity and future-proof referencing.
[Rev] Corrected.
Comments 3.6:
Page 3. Add range of diagnostic accuracy for INSM1 (Recent studies have suggested high diagnostic accuracy…[...]).
Response 3.6:
Thank you for your comment. As suggested, we have added the reported range of diagnostic accuracy for INSM1 to the penultimate paragraph of the Discussion section (page 3). Specifically, we now state that recent studies have suggested high diagnostic accuracy of INSM1 in identifying pulmonary neuroendocrine tumors, particularly SCLC and LCNEC, with reported sensitivity ranging from 68% to 95% and specificity from 95% to 99%.
[Rev] Corrected.
Comments 3.7:
Methods
Study Search
Page 3. Fix wrong numbering of Supplementary table: Supplementary Table 1. Search strategy for systematic review, while in-text is pointing to table S2 “are provided as supplementary material. (Table S2).”
Response 3.7:
Thank you for pointing this out. We have corrected the mismatch between the in-text citation and the Supplementary Table numbering. The reference to " Supplementary Table 1" in the main text now correctly reads " Supplementary Table 2" to match the actual table in the supplementary (page 3).
[Rev] Supplementary Table 1 does not exist; perhaps consider renumbering Supplementary Table 2.
Comments 3.8:
Page 4. I would add a citation to QUADAS-2 on the first instance it occurs, i.e., “Quality Assessment of Diagnostic Accuracy Studies-2 (QUADAS-2) evaluation.”
Response 3.8:
Thank you for your comment. As suggested, we have added the appropriate citation for QUADAS-2 at its first mention in the manuscript (i.e., “Quality Assessment of Diagnostic Accuracy Studies-2 [QUADAS-2] evaluation”) (page 4).
[Rev] Corrected.
Comments 3.9:
Use of Generative AI Tools
Page 4. Missing figure number in the sentence, i.e., “The figure outlines the two diagnostic models assessed in this study.”
Response 3.9:
Thank you for your comment. We would like to clarify that the sentence “The figure outlines the two diagnostic models assessed in this study.” appears as part of the figure caption, not in the main text. The corresponding models are already referenced in the main text as “Two analytical models were employed (Table 1):” in the first sentence of the Outcomes subsection (page 4).
However, we noted that Table 1 was previously inserted in a less appropriate location, so we have repositioned it immediately below the relevant paragraph to improve clarity and alignment with its first in-text citation.
[Rev] Corrected.
Comments 3.10:
Outcomes
Redo this section altogether with table 1, since it is highly confusing.
Response 3.10:
Thank you for your comment. In response, we have fully revised the Outcomes subsection and Table 1 (page 4-5) to improve clarity. To enhance visual distinction, we applied shading to the table: the positive group (LCNEC and SCLC) is now shown in dark gray, while the negative group (NSCLC other than LCNEC) is shown in light gray.
We also recognize that the relationship between LCNEC and NSCLC may have been unclear. To address this, we have revised the figure to better illustrate that LCNEC is a subtype of NSCLC, but is classified as a neuroendocrine tumors that typically expresses INSM1, and was therefore grouped as INSM1-positive in our analysis.
[Rev] Corrected.
Comments 3.11:
Results
Page 5. Misspelled results: “RESLUTS”
Response 3.11:
Thank you very much for pointing out. We corrected the spelling as suggested(page 6).
[Rev] Corrected.
Comments 3.12:
Page 6. Check journal policies regarding abbreviations position in the manuscript, but shouldn’t go in this section, “All abbreviations are defined as follows: INSM1, insulinoma-associated protein 1; LCNEC, large cell neuroendocrine carcinoma; WOS, Web of Science.”
Response 3.12:
Thank you for your comment. We would like to clarify that the list of abbreviations appears in the figure legend for Figure 1, not in the main text. To avoid confusion, we have ensured that all abbreviations used within each figure or table are defined immediately following the corresponding figure or table legend. We have removed any standalone abbreviation lists from inappropriate sections of the text and have placed all definitions in close proximity to where the abbreviations are actually used.
[Rev] Corrected.
Comments 3.13:
Page 6, table 2. Spell out the meaning of “ANS”, “CB”, “SG”, and all other abbreviations used.
Response 3.13:
Thank you for your comment. We have addressed the abbreviation issues in Table 2 as follows:
“ANS” has been retained and is now defined as “any nuclear staining” in the abbreviation list.
“CB” (cell block) referred to pleural effusion specimens and has been replaced with the more descriptive term “pleural effusion” to improve clarity.
“SG” (surgical specimen) has been replaced with “surg”, which is used consistently throughout the table.
[Rev] Corrected.
Comments 3.14:
Page 8-9, figure 3-4. Improve figure caption by describing the meaning of elements of the forest plot such as: dot size meaning; shapes (circle, diamond), line length for each study.
Response 3.14:
Thank you for your comment. To improve clarity, we have revised the figure captions for both Figure 3 and Figure 4 (page 9-10). Specifically, we added explanations for the elements of the forest plots, including the meaning of dot size, diamond shapes, and horizontal lines representing the 95% confidence intervals. For the SROC curves, we clarified that each dot represents an individual study plotted in ROC space (sensitivity vs. 1 − specificity).
[Rev] Corrected.
Comments 3.15:
Page 8-9, figure 3-4. Generate confidence bands at 95% for the SROC curves (C).
Response 3.15:
Thank you for your comment. Unfortunately, our analysis software (Meta-DiSc1.4 and mada in R) does not support the generation of 95% confidence bands for SROC curves. While we acknowledge the value of displaying such intervals, we have instead provided pooled estimates with 95% confidence intervals for sensitivity, specificity, and diagnostic odds ratios, which reflect the uncertainty around diagnostic performance. We hope this provides sufficient statistical context for interpretation.
The DOR values were newly added in response to comments from another reviewer. Estimates of the diagnostic odds ratio (DOR) were derived using a random-effects model in Meta-DiSc version 1.4, and the results have been added to the third paragraph of both the Diagnostic Accuracy of INSM1 in the NSCLC Model (page 8) and the Diagnostic Accuracy of INSM1 in the Lung Cancer Model (page 9) subsections in the Results section.
[Rev] Corrected.
Comments 3.16:
Discussion
Include the absence of an assessment for publication.
Response 3.16
Thank you very much for your thoughtful recommendation. It is well recognized that assessing publication bias in diagnostic test accuracy (DTA) meta-analyses is more challenging than in intervention reviews due to several methodological complexities. These include typically high diagnostic odds ratios (DORs), variation in positivity thresholds across studies, and imbalanced numbers of diseased and non-diseased participants—all of which affect precision and funnel plot symmetry, as discussed by van Enstet al., Investigation of publication bias in meta-analyses of diagnostic test accuracy: a meta-epidemiological study (BMC Med Res Methodol. 2014;14:70).
While Deeks’ test is recommended for assessing publication bias in DTA meta-analyses, it has limited power in the presence of heterogeneity—an inherent characteristic of DTA studies—and therefore its interpretation is limited.
Based on the above considerations, we did not perform publication bias analysis in our bivariate DTA meta-analysis. We have also added statements explaining this decision and the rationale behind it as the fourth limitation of this study, in the penultimate paragraph of the Discussion section (page 13).
[Rev] Corrected.